# Australia’s Role in Pneumococcal and Human Papillomavirus Vaccine Evaluation in Asia-Pacific

**DOI:** 10.3390/vaccines9080921

**Published:** 2021-08-18

**Authors:** Zheng Quan Toh, Chau Quang, Joseph A. Tooma, Suzanne M. Garland, Kim Mulholland, Paul V. Licciardi

**Affiliations:** 1Murdoch Children’s Research Institute, Parkville, VIC 3052, Australia; zhenq.quantoh@mcri.edu.au (Z.Q.T.); chau.quang@mcri.edu.au (C.Q.); Suzanne.garland@thewomens.org.au (S.M.G.); kim.mulholland@mcri.edu.au (K.M.); 2Department of Paediatrics, The University of Melbourne, Parkville, VIC 3052, Australia; 3Australia Cervical Cancer Foundation, Fortitude Valley, QLD 4006, Australia; joe.tooma@accf.org.au; 4Department of Obstetrics and Gynaecology, University of Melbourne, Parkville, VIC 3052, Australia; 5Regional WHO HPV Reference Laboratory, Centre Women’s Infectious Diseases Research, The Royal Women’s Hospital, Parkville, VIC 3052, Australia; 6Department of Infectious Disease Epidemiology, London School of Hygiene and Tropical Medicine, London WC1E 7HT, UK

**Keywords:** alternative vaccine schedules, pneumococcal vaccines, human papillomavirus vaccine, vaccine evaluation and Australia

## Abstract

Australian researchers have made substantial contributions to the field of vaccinology over many decades. Two examples of this contribution relate to pneumococcal vaccines and the human papillomavirus (HPV) vaccine, with a focus on improving access to these vaccines in low- and lower-middle-income countries (LLMICs). Many LLMICs considering introducing one or both of these vaccines into their National Immunisation Programs face significant barriers such as cost, logistics associated with vaccine delivery. These countries also often lack the resources and expertise to undertake the necessary studies to evaluate vaccine performance. This review summarizes the role of Australia in the development and/or evaluation of pneumococcal vaccines and the HPV vaccine, including the use of alternative vaccine strategies among countries situated in the Asia-Pacific region. The outcomes of these research programs have had significant global health impacts, highlighting the importance of these vaccines in preventing pneumococcal disease as well as HPV-associated diseases.

## 1. Introduction

The pneumococcal vaccine and the human papillomavirus (HPV) vaccine are both highly successful vaccines in terms of reducing invasive and non-invasive pneumococcal disease (IPD) and HPV-associated diseases, respectively. Both vaccines are very costly, which means that many low- and lower-middle-income countries (LLMICs) are unable to afford introducing or sustaining these vaccines into their National Immunisation Programs (NIPs) without support from international organizations (i.e., PATH, the Bill and Melinda Gates Foundation, and Gavi, the Vaccine Alliance) [1,2]. Alternate vaccine schedules such as a reduced number of doses and/or extended durations between doses would alleviate cost and logistical difficulties associated with vaccine implementation in LLMICs, provided that similar vaccine efficacy and/or immunogenicity can be demonstrated between the original schedule and alternate schedules in randomised controlled trials. Evaluation of alternative dose schedules has been an important part of the global pneumococcal vaccine and HPV vaccine research agenda, in which Australian researchers have had a leading role.

Measuring vaccine impact in LLMICs once they have introduced these vaccines is also critical to inform health policies. While high-income countries typically have appropriate data collection systems and databases to monitor vaccine coverage, vaccine impact and effectiveness (including surveillance indicators), as well as vaccine safety, these systems are often lacking in many LLMICs [2]. There are many LLMICs in the Asia-Pacific region that have not yet introduced the pneumococcal vaccine and/or the HPV vaccine into their NIPs or have only just introduced them in the last 10 years. Many of these countries have limited resources and experience in monitoring the introduction and impact of new vaccines. In this review, we highlight Australia’s role in measuring the impacts of the pneumococcal and HPV vaccine as well as evaluating alternative schedules among countries in the Asia-Pacific region, thus contributing to global health impact.

## 2. Pneumococcal Vaccine

### 2.1. Burden of Disease and Pneumococcal Conjugate Vaccine

*Streptococcus pneumoniae*, also known as the pneumococcus, is a major cause of bacterial pneumonia in the young, the elderly, and immunocompromised individuals [3]. Between 2000 and 2015, it is estimated that >300,000 deaths in children aged 1–59 months were caused by pneumococcus [4]. Most of these deaths occurred in LLMICs due to the inaccessibility of pneumococcal vaccines, owing to their high cost, and limited access to disease treatment (i.e., oxygen and/or antimicrobial therapy) [5].

There are two types of pneumococcal vaccines available (Table 1); the pneumococcal polysaccharide vaccine (PPV) and the pneumococcal conjugate vaccine (PCV). PPV has been recommended for use in older children (>2 years old) and adults who are at increased risk of pneumococcal diseases [6], while PCV is recommended for children <2 years of age as well as in older adults. There is a long history of Australian-led research on both the use of PPV and PCV in LLMICs and high burden settings (i.e., Indigenous Australians).

### 2.2. Pneumococcal Polysaccharide Vaccine

In the 1970s, Australian researchers conducted important vaccine trials in Papua New Guinea (with high pneumococcal disease burden), where the findings were crucial to the subsequent licensure of the first pneumococcal vaccine (23vPPV) in 1983 [7] (Table 2). In particular, the study found that vaccination of adults with a 14-valent PPV (first generation PPV) reduced pneumococcal infection and death by 81% and 44%, respectively [8]. While PPV was effective in adults, there was little impact in children, particularly those <2 years old [9]. The low efficacy of PPV in children was attributable to the low immunogenicity of the vaccine to some serotypes in children (as a result of an immature immune system), including those serotypes that are commonly responsible for invasive diseases in children [10]. These findings, along with others, contributed to the recommendations for the use PPV in children >2 years of age.

In high pneumococcal burden settings such as Papua New Guinea and among Indigenous Australians, young infants can be colonised with pneumococcus within their first month of life, and it represents the highest risk for IPD [21,22,23]. Maternal immunisation is one way to prevent early pneumococcal carriage in young infants, and this protection is thought to be mediated through antibody transfer at the time of delivery and/or breast feeding. Lehmann et al. found that serotype-specific antibodies (5 and 23F) were significantly higher in children of immunised women than unimmunised women for up to age 2 months and for up to age 4 months for serotype 14, providing evidence that such intervention prevents early pneumococcal carriage [16]. This study, along with two other studies in Gambia [24] and Bangladesh [25], contributed to the earliest data on maternal immunisation.

### 2.3. Pneumococcal Conjugate Vaccine

Countries that introduced PCV, including Australia, have observed large reductions in pneumonia and IPD [4]. Of the 73 Gavi-eligible countries, 59 (81%) introduced PCV into their NIPs. In contrast, only about 50% of lower- and upper-middle income countries (non-Gavi eligible) introduced PCV into their NIPs as these countries often do not have the support from international health-care organizations, as do low-income countries [26]. Most of these countries that are yet to introduce PCV are in Asia and Africa, where disease burden is highest, and include countries with large populations, such as China and Vietnam [27]. In countries that can afford to introduce PCV into NIPs, serotype replacement is a major concern, since currently available PCVs cover only 10–13 of the >100 pneumococcal serotypes. The extent to which replacement occurs in LLMIC, where the burden of disease is the highest, is unknown. Australian researchers have contributed to evaluating the optimal pneumococcal vaccine schedules for LLMICs, as well as monitoring the impact of pneumococcal vaccines in LLMICs and in high-risk communities (e.g., the Australian Indigenous population), as discussed below.

### 2.4. Alternative Pneumococcal Vaccine Schedules

The cost of the PCV is a barrier for its use and sustainability [28]. A three-dose schedule is currently recommended by the WHO in children. Using fewer PCV doses, such as one primary dose with one booster dose (1 + 1 schedule), could be a more cost-effective way of using this vaccine to maintain herd protection and may improve the vaccine’s financial sustainability [29]. Other strategies to improve protection in high-burden settings include maternal immunisation, as well as combining different PCVs (PCV10 and PCV13) to broaden protection against respiratory pathogens such as pneumococci and non-typeable *Haemophilus influenzae* (NTHi).

Table 3 summarises studies in which Australian researchers have played a crucial role in evaluating alternative pneumococcal vaccination schedules for LLMICs and in other high burden settings. These studies had a major global health impact. For example, the study conducted in Fiji by Russell et al. led to the introduction of PCV into Fiji’s NIP (3 + 0 schedule) in 2012 [30,31]. The study was also the first to show that a single dose of PCV7 given during infancy may offer some protection for most vaccine serotypes. Other important research relevant for high burden settings (i.e., Papua New Guinea), where pneumococcal colonisation occurs very early in life, relates to the evaluation of neonatal pneumococcal vaccination. This strategy was demonstrated to be safe and immunogenic [11,12,32]. Higher valency vaccine may also be needed in high burden settings, since infants can be colonised by multiple serotypes, including those not included in current PCVs [11,12]. An alternative strategy to increase serotype coverage until new PCVs with broader serotypes, or serotype-independent vaccines, become available, is the combination of priming with three doses of PCV and boosting with one dose of 23vPPV [33]. This strategy, however, was found to induce short-term immune hypo-responsiveness, although the clinical significance is unknown [34,35,36].

### 2.5. Pneumococcal Vaccine Impact

Evaluation of the distribution of pneumococcal serotypes causing invasive diseases and/or carriage prior to vaccine introduction is crucial for measuring vaccine impact and serotype replacement. Researchers from Australia have been involved in pneumococcal vaccine evaluation studies in the Asia-Pacific region (Table 4). The studies in Table 4 demonstrated significant reductions in vaccine-serotype carriage and hospital admission due to pneumococcal pneumonia and/or acute lower respiratory infections (ALRIs) following the introduction of the pneumococcal vaccine, with some exceptions in Australian Indigenous cohort studies. Early effectiveness studies in Australian Indigenous infants found an increased risk of ALRIs, no change in otitis media incidence and radiologically confirmed pneumonia following three primary doses of PCV7 and a dose of 23vPPV at 18 months [48,49,50]. It was hypothesized that this limited vaccine impact could be due to early pneumococcal carriage, carriage of non-vaccine serotypes responsible for severe pneumonia, non-vaccine serotype/other respiratory pathogen replacement in the respiratory tract, as well as the immune hypo-responsiveness associated with 23vPPV. Findings from this study led to the revision of PCV immunisation schedules for Australian Indigenous infants (removal of 23vPPV at 18 months) [51]. Indirect effects on vaccine-serotype carriage in adults have been documented following PCV vaccine introduction. Not surprisingly, there has been an increase in non-vaccine serotypes in some countries (Fiji and Mongolia where data is available). The extent of serotype replacement in LLMICs, particularly countries in the Asia-Pacific region, is poorly understood, and threatens the control of pneumococcal disease. This knowledge gap highlights the need for continued surveillance and monitoring of vaccine impact in the region.

## 3. Human Papillomavirus Vaccine

### 3.1. Burden of Disease and HPV Vaccines

HPV is a broad group of viruses with more than 200 genotypes, some of which have tropism for skin, and others (~30 to 40 genotypes) for the genital mucosal and skin area [63]. Genital HPVs are transmitted by close contact, between genital skin and genital skin/mucosa, and are the most common viral sexually transmitted infection. It is estimated that approximately 80% of sexually active individuals will be infected by HPV at some stage in life, especially early after sexual debut [64]. HPV is known to cause a range of diseases from anogenital warts and benign/low-grade genital abnormalities (the viral cytopathic response) to invasive anogenital cancers, particularly cervical cancer [63]. Cervical cancer is the fourth most common cancer in women worldwide, with 604,000 cases and 342,000 deaths in 2020 [65]. There are 20 HPV genotypes that are known to cause cancer (oncogenic of high-risk types), with HPV 16 and 18 together accounting for 70% of cervical cancers worldwide [66,67].

HPV was first identified to be the causal agent of cervical cancer in the early 1980s by Harald zur Hausen and his team [68]. Since then, there has been extensive research into the prevention of cervical cancer, including the prevention of HPV infection through vaccination. The breakthrough in the development of the HPV vaccine was the discovery of the self-assembly capsid viral proteins into virus-like particles (VLPs) in Australia, and also elsewhere by others [69]. This technology subsequently became the basis for the current prophylactic HPV vaccines.

There are currently three licensed prophylactic HPV vaccines and one under review for WHO pre-qualification (Table 5). These vaccines are highly immunogenic and effective in preventing vaccine-type HPV infection, cervical pre-cancers, and cancers [70,71]. Both 4vHPV and 9vHPV are also effective against genital warts, and other vaccine-type anogenital pre-cancers such as vulvar, vaginal, and anal [72,73].

### 3.2. Australian HPV Vaccine Program and Impact

Australia was one of the first countries to introduce a government-funded school-based HPV vaccine program (4vHPV in 2007). The program was first introduced as a female only program and achieved high vaccine coverage (around 80% for three doses) in women <18 years of age. The vaccine impact was one of the first reported globally [74] (summarised in Table 6). Within the first five years of HPV vaccine introduction, significant decreases in vaccine-type HPV prevalence were observed in both men (as a result of herd protection from the female only program at the time) and women (also with herd protection of same age vaccine eligible women), as well as high-grade cervical abnormalities in women [75,76,77]. The prevalence of high-risk vaccine-type HPV declined from 22% in the pre-vaccine era to 1.5% among girls aged 18–24 years old, within nine years following introduction of the vaccine [76,78]. Cross protection against closely related HPV vaccine types (HPV 31/33/45), as represented by a decrease in HPV genotype prevalence, was also observed six years after introduction of the vaccine [77]. Seven-years post-4vHPV-introduction in Australia, a national data linkage analysis reported 40% vaccine effectiveness against high-grade cervical intraepithelial neoplasia (CIN) (all cause, non-HPV type specific, histologically confirmed) [79]. The vaccine was effective against CIN, regardless of whether women have received one, two or three doses of 4vHPV, suggesting that one dose of HPV vaccine may be sufficient for protection. Single dose HPV vaccine schedules are particularly relevant for LLMICs, where high costs and logistical difficulties in vaccine delivery are major barriers to vaccine implementation.

In 2013, Australia introduced a gender-neutral HPV vaccination program. The benefits of gender-neutral HPV vaccination include direct protection for men (including men-who-have-sex-with-men (MSM), who do not benefit from female only vaccination) and the provision of herd protection for unvaccinated women [80]. In many high-income countries where cervical cancer is controlled by vaccination and cervical cancer screening, the risk of anal cancer for MSM can be as high as the risk of cervical cancer for women [81], highlighting the importance of HPV vaccination in boys and young men. Chow et al. recently reported on a repeat cross-sectional study conducted in MSM comparing the HPV prevalence before and after the introduction of gender-neutral HPV vaccination in Australia [82]. They found a significant reduction in the prevalence of HPV genotypes 6, 11, 16, or 18 in the anus (76%), penis (52%), and oral cavity (90%) compared with a pre-vaccination cohort, demonstrating the first direct impact of HPV prevalence in MSM after the implementation of the gender-neutral HPV vaccination programme [82], which is likely to lead to reductions in anal cancer incidence.

In 2018, 9vHPV was introduced as a two-dose schedule to replace 4vHPV in Australia. It was postulated that the replacement of 4vHPV with 9vHPV in Australia will protect against an additional 15% and 11% of cervical cancer and anal cancers, respectively [74]. With the use of 9vHPV coupled with high vaccine coverage in a gender-neutral vaccination program, and robust HPV cervical screening, Australia is likely to be the first country to eliminate cervical cancer (as defined as <4 new cases per 100,000 women each year) by 2028 [83]. The incidence of cervical cancer is expected to further decrease to <1 case per 100,000 women by 2066 [83]. Indeed, completely vaccinated women in Australia were found to have less than half the incidence rate of cervical intraepithelial neoplasia grade 3 and/or cervical adenocarcinoma in Situ than in unvaccinated women (2.8 cases compared with 6.0 cases per 1000 women). A trend of lower incidence of cervical cancer in HPV-vaccinated than in HPV-unvaccinated women was also reported, although longer follow up data are required to verify this observation [84].

### 3.3. Alternate HPV Vaccine Strategies

Licensed HPV vaccines were originally given as a three-dose schedule. In 2014, the WHO recommended a two-dose schedule (six months apart) for girls/boys <15 years old [91]. This was based on immune-bridging studies that demonstrated non-inferior antibody levels in girls <15 years old who received two doses compared to older women aged 16–26 years old who received three doses (where efficacies against HPV infection and cervical pre-cancer have been established) [92].

Australia researchers were involved in the evaluation of reduced-dose HPV vaccine schedules for Fiji and Mongolia. A cohort study in Fiji found that girls who received two doses of 4vHPV had similar immune responses, after six years, to girls who received the standard three-dose schedule [93,94,95]. This was the longest follow up of reduced-dose schedules at the time, supporting the WHO recommendation of a two-dose schedule. More interestingly, girls who received a single dose of HPV vaccine had higher antibody levels than unvaccinated girls (albeit lower levels than those who received two or three doses) after six years, and these were boosted to a similar level as girls who received two or three doses following a booster dose of 2vHPV [94]. This was the first study to demonstrate the generation of immunological memory following just one dose of HPV vaccine, as well as the first to report on the immunogenicity following a mixed vaccine schedule [94]. These findings, along with other studies on the single dose schedule [96,97,98,99], supported further research into this field [100,101], leading to several ongoing Phase III clinical trials [98,102,103]. It is, however, important to note that the clinical relevance of lower antibodies generated following one dose of HPV vaccine is unknown since there is no identified immune correlate of protection. Recent data from Mongolia which has one of the highest cervical cancer rates in Asia (age-standardised rate of 19.7/100,000) [104], reported 92% reduction in the prevalence of vaccine-type HPV 16 and 18 in girls who only received one dose of 4vHPV six years earlier compared with unvaccinated girls [105]. In addition, 90% and 58% of vaccinated women remained seropositive for HPV 16 and 18, respectively, with antibody levels significantly higher than unvaccinated women [105]. This study not only contributes to the limited HPV vaccine research in Mongolia, but also the emerging evidence of single-dose HPV vaccine schedules globally. A single dose schedule will alleviate the constraints (high vaccine costs and difficulties in vaccine delivery) faced by many LLMICs, where the burden of cervical cancer is the highest.

Another alternate strategy is to vaccinate those that are at the highest risk of HPV infection and cervical cancer, such as those with high numbers of sexual partners, in addition to cervical cancer screening. While HPV vaccines are prophylactic and do not clear existing lesions [106], it is increasingly being recognised that there are still important benefits for vaccinating HPV-infected women [107], particularly in settings where cervical cancer screening is limited (i.e., LLMICs and remote settings). These benefits include reducing transmission, protecting against vaccine-type HPV that the individual is not infected with, as well as reducing the risks of clinical disease relapse after treatment. This concept is particularly relevant for female sex workers (FSWs) who have a very high risk of HPV infection and cervical cancer, due to the high number of sexual partners [108]. It is also common for them to harbour multiple HPV genotype infections [109], potentially serving as a reservoir for transmitting HPV within the community. We are conducting a pilot study in Vietnam to investigate a targeted HPV vaccination strategy towards FSWs to reduce their risk of HPV infection and protect them against HPV-associated diseases, as well as reducing HPV transmission within the community [110].

### 3.4. HPV Vaccine Introduction in LLMICs

The contribution of Australian researchers to global HPV vaccine research has been fundamental to the introduction of HPV vaccines in the Asia-Pacific region. In addition, the Australian Cervical Cancer Foundation (ACCF) in partnership with either Gavi or Gardasil Access has been involved in a number of HPV vaccine demonstration projects and HPV educational programs over the past decade in LLMICs including Nepal, Bhutan, Kiribati, Vanuatu, the Solomon Islands and Papua New Guinea [111]. In particular, the ACCF has been successful in facilitating the introduction of the HPV vaccine in Bhutan, the first developing country to have a national HPV vaccination program and, more recently, in the Solomon Islands in 2019. As of October 2018, the ACCF has contributed more than 500,000 doses of HPV vaccine to these countries and has established sufficient local health authority capacity to be able to successfully undertake a complex HPV vaccination program (J. Tooma, personal communication).

## 4. Conclusions

Australia has made substantial contributions to the measurement of the impact of pneumococcal and HPV vaccines, as well as facilitating the introduction of the HPV vaccine in LLMICs. These have important implications in providing real world evidence that the vaccines are effective, and in sharing the lessons learnt both at the country level where the research was conducted, as well as for the global health communities. Ongoing research in these areas will provide the necessary justification and steps for countries that have yet to introduce pneumococcal vaccines and/or the HPV vaccine to do so, as well as to inform governments on the impact of these vaccines, so that other health funds can be directed to diseases that are not preventable. Alternative schedules that alleviate high vaccine costs and logistical constraints in vaccine delivery will improve vaccine access and reduce inequality in LLMICs. Current PCV evidence suggests that two primary doses separated by at least two months followed by a later booster dose at or after 9 months of age would provide protection for children in high burden settings, while a single-dose HPV vaccination schedule has shown encouraging results. Randomized controlled trials are ongoing to evaluate these schedules and, if successful, will significantly improve vaccine access in LLMICs that have yet to introduce these vaccines.

## Figures and Tables

**Table 1 vaccines-09-00921-t001:** Pneumococcal vaccines.

Manufacturer	Merck Sharp & Dohme Corp (23vPPV)	Wyeth (Prevenar, PCV7) *	GSK (Synflorix, PCV10)	Pfizer (PCV13)	Serum Institute of India (Pneumosil, pPCV10)
Year licensed	1983	2000	2009	2010	WHO prequalified 2020
Common serotypes	6B, 9V, 14, 19F and 23F				
Additional serotypes	1, 2, 3, 4, 5, 7F, 8, 9N, 10A, 11A, 12F, 15B, 17F, 18C, 19A, 20, 22F, 33F	4, 18C	1, 4, 5, 7F, 18C,	1, 3, 4, 5, 6A, 7F, 18C, 19A,	1, 5, 6A, 7F, 19A,
Carrier protein (s)	-	Non-toxic diphtheria CRM197	^ NTHi Protein D- tetanus toxoid- diphtheria toxoid	Non-toxic diphtheria CRM197	Non-toxic diphtheria CRM197

CRM: cross-reactive material; NTHi: Non-Typeable *Haemophilus influenzae*; * replaced by PCV13; ^ Serotypes 1, 4, 5, 6B, 7F, 9V, 14 and 23F conjugated to NTHi Protein D; serotype 18C conjugated to tetanus toxoid; serotype 19F conjugated to diphtheria toxoid.

**Table 2 vaccines-09-00921-t002:** Pneumococcal polysaccharide vaccine evaluation studies.

Country/Population. Study Year (s)	Primary Aim	Study Design	Main Findings	Reference
Papua New Guinea (Southern Highlands Province), 1970	To determine the efficacy of a 14vPPV in adults	Adults (*n* = 11,958, >10 years old) were randomised to receive 14vPPV or placebo (saline) A subset of participants was followed up for clinical, immunology and bacteriology assessments	Compared to placebo, vaccinated group had significantly lower pneumococci in blood-culture and/or lung aspirates (*n* = 136, 14 placebo vs. 2 in PPV) Among 303 deaths, 41 placebo vs. 23 in PPV were due to pneumonia	[8]
Papua New Guinea (Southern Highlands Province), 1972–1973	To evaluate the efficacy of a 14vPPV in children	Children aged between 6 months and 5 years (*n* = 871) were randomised to receive 14vPPV or placebo (saline)	Vaccinated children aged ≥ 17 months had 37% lower incidence of ALRI compared to placebo (74 vs. 39 cases); no protection for children <16 months old. 8 deaths from ALRI in the placebo group, compared to only 1 death in the vaccine group.	[11]
Papua New Guinea (Eastern and Southern Highlands Provinces), 1981	To determine the efficacy of a 14vPPV in children; combined analysis from 3 trials	Children aged between 6 months and 5 years (*n* = 871, *n* = 1487, *n* = 4862) were randomised to receive 14vPPV or placebo (saline). A subset of participants received a second dose at after 12 months	Vaccine efficacy for children vaccinated at <6 months, <2 years, and all ages for death from ALRI was 31 %, 50%, and 59%. Mortality from all causes was 19% less in the vaccinated group compared to placebo.	[12]
Australia (Non-indigenous), 1980	To determine the efficacy of a 14vPPV in children	Children aged 6 to 54 months (*n* = 1273) were randomised to receive 14vPPV (*n* = 634) or placebo (saline)(*n* = 639) Different dosage regimens were used for children <2 and ≥2 years old; approximately half of children <2 years received a booster dose at 6 months	No statistically significant difference between placebo and vaccine recipients for hospitalisation due to respiratory tract infections and/or otitis associated disease and/or vaccine-type carriage. Antibody response varied with the age of the child and was serotype dependent. Antibody response to serotypes (6A, 14, 19F, and 23F) commonly associated with pneumococcal disease in childhood were low up until 5 years of age. Booster dose given at 6 months after the primary dose, were not associated with significant increase in serum antibody levels for 6A, 23F, 19F and 14.	[10,13,14]
Australia (Indigenous cohort), 1982	To determine the immunogenicity and vaccine efficacy of 23vPPV against respiratory infections and carriage in Indigenous Australian children	Children aged 6 months to 5 years (*n* = 60) were randomised to receive 14vPPV (*n* = 30) or placebo (saline) (*n* = 30) Different dosage regimen were used for children <2 and ≥2 years old	Indigenous participants produced lower antibody levels pre-(baseline) and post-PPV compared to non-indigenous cohort (above) despite higher vaccine type carriage in Indigenous population. No differences in new otitis media episodes between PPV and placebo groups. PPV group had reduced vaccine-type carriage but were not statistically significant; children <2 years old carriage rates returned to pre-immunisation levels after 3 months while children >2 years old remained low for 9 months.	[15]
Papua New Guinea (Southern Highlands Province), 1991–1994	To determine the immunogenicity of a 23vPPV given to pregnant women and transfer of pneumococcal antibody and the persistence of pneumococcal antibody in early infancy	Women at 28–38 weeks gestation were recruited to receive a dose of 23vPPV, and their child was followed up for serology testing to serotypes 5, 7F, 14 and 23. An unvaccinated control group of women who had not received the vaccine and their child was also recruited.	Significant increase in pneumococcal antibody titres following vaccination for serotypes 5, 14 and 23F but not 7F. Vaccinated mothers and cord samples have significantly higher antibody response to all 4 serotypes when compared to unvaccinated. Antibody titres in infants declined rapidly after birth to a level that were similar to unvaccinated group after 2 months except for serotype 14, which remains higher in vaccinated group up to 4 months.	[16]
Australia (Indigenous cohort), 2006–2011	To determine the impact of the 23vPPV in pregnant women against infant middle ear disease, pneumococcal carriage and ALRI	Healthy Indigenous women aged 17–39 years were randomised to receive the 23vPPV: - during pregnancy (*n* = 75; 30–36 weeks gestation); - at birth (*n* = 75); or - at 7 months post-partum (*n* = 77).	Low vaccine efficacy against infant ear disease (12%, 95% CI 12% to 31%) and 23vPPV-type carriage (30%, 95% CI−34% to 64%). Antenatal 23vPPV vaccination was not associated with a reduced incidence of infant ALRI hospitalisations or clinic presentations during the first year of life.	[17,18]
Australia (Indigenous cohort), 2010–2011	To evaluate the immune response to a first and second dose of 23vPPV in Indigenous adults and a first dose of 23vPPV in non-Indigenous adults	Adults aged 15–59 years in remote Indigenous communities, who are due for a first or second dose of 23vPPV Group 1: first dose of 23vPPV in Indigenous adults (*n* = 60) Group 2: second dose of 23vPPV in Indigenous adults (*n* = 20) Group 3: first dose of 23vPPV in non-Indigenous adults (*n* = 25)	Group 1 and 3 had higher post-vaccination serotype-specific IgG levels for most vaccine serotypes than Group 2. Group 3 had significantly higher median adequate response (to 21/23 serotypes) than Group 1 (18/23 serotypes) and 2 (15/23 serotypes); no significant difference between Group 1 and 2. No significant differences in post-vaccination serotype-specific IgG or OI levels or memory B cell numbers between all groups, except Group 3 had significantly higher IgG and OI levels for serotype 1 and higher number of memory B cells for serotype 6B.	[19,20]

14vPPV: first generation 14-valent PPV (Merck Sharpe & Dohme) containing serotype 1, 2, 3, 4, 5, 6, 7, 8, 12, 14, 18, 23, 25, and 46; ALRI: Acute lower respiratory infection.

**Table 3 vaccines-09-00921-t003:** Evaluation of alternative pneumococcal vaccine schedules.

Country, Study Year (s)	Primary Aim	Study Design	Main Findings	Reference
Fiji, 2003–2008	To determine the optimal pneumococcal vaccine schedule	Healthy infants randomised to 3 PCV7 groups (*n* = 552): - 1 dose (aged 6 weeks) - 2 doses (aged 6 and 14 weeks) - 3 doses (aged 6, 10, and 14 weeks) Half of each group received 23vPPV at age 12 months. All received a small dose of 23vPPV at 17 months	Three dose PCV schedule is more immunogenic than 1 or 2 doses Priming with a single dose was better than 2 or 3 doses. Less vaccine-type pneumococcal carriage with more doses of PCV Children who did not receive the 23vPPV had significantly higher antibody levels for all PCV serotypes compared with those who received 23vPPV following exposure to a small dose of 23vPPV (immune hyporesponsiveness)	[30,34,36,37,38]
Papua New Guinea, 2005–2009	To determine the impact of early schedules on pneumococcal carriage	Healthy infants randomized at birth to receive PCV7 in a - 0–1–2-month (*n* = 101) or - 1–2–3-month (*n* = 105) or - no vaccine (control, *n* = 106). All children received 23vPPV at age 9 months	PCV recipients had higher antibody response for all vaccine types except 6B than controls at age 3 months PPV induced significantly higher vaccine-type antibody responses in PCV7-primed infants than in controls No significant differences in PCV7 serotype carriage between PCV7 recipients and controls at any age Prevalence of non-PCV7 carriage was 17% higher in 7vPCV recipients (48%) than in controls (25%) at 9 months of age.	[39,40]
Australia (Indigenous cohort), 2011–2017	To determine if combination of PCV vaccines is superior to single vaccine schedules against otitis media pathogens	Three randomised groups (1:1:1) (*n* = 425): (1) PCV10 at 2, 4, 6 months of age; (2) PCV13 at 2, 4 and 6 months of age; (3) investigational schedule: PCV at 1, 2 and 4 months plus PCV13 at 6 months of age.	A combined schedule of PCV10 and PCV13 at 1–2–4–6 months is safe and immunogenic against PCV13 serotypes and protein D One dose of PCV10 at 1-month is immunogenic A 4-dose schedule is superior to either 3-dose schedule of PCV10/13 No significant differences in prevalence of any form of otitis media between vaccine groups at any age	[41,42,43]
Papua New Guinea, 2011–2014	To compare PCV10 and PCV13	Three doses of PCV10 vs. 3 doses of PCV13 (1, 2 and 3 months of age), *n* = 262 Children were randomised at 9 months to receive 23vPPV or no 23vPPV All received a micro-dose of PPV at 23 months	Both PCV10 and PCV13 were immunogenic and well tolerated More than 90% had seroprotective antibody levels against most vaccine serotypes at 4 months of age Carriage of any shared PCV10/13 serotypes were similar between the groups at 4 or at 9 months of age Significant increase in IgG responses for all 23vPPV-serotypes at 10 months of age post-23vPPV 23vPPV induced high levels of seroprotection when given under the age of 12 months to PCV-primed children in high-risk settings Both PPV-vaccinated and PPV-naive children produced IgG antibody above the seroprotective titer to a micro dose of PPV	[21,33]
Vietnam (Nha Trang), 2016-ongoing	To evaluate different PCV schedules	RCT (*n* = 45360): (1) PCV10: 3 + 0 (2, 3, 4 months) (2) PCV10: 2 + 1 (2, 4, 12 months) (3) PCV10: 1 + 1 (2, 12 months) (4) PCV10: 0 + 1 (12 month)	Ongoing study. Primary outcome: Vaccine type pneumococcal carriage among children receiving different PCV schedules Non-inferiority between 1 + 1 group and those receiving 2 + 1/3 + 0 groups.	[44]
Vietnam (Ho Chi Minh City), 2013–2015	To evaluate different PCV schedules and to provide a head-to-head comparison of PCV10 and PCV13	RCT (*n* = 1400): (1) PCV10: 3 + 1 (2, 3, 4, 9 months) (2) PCV10: 3 + 0 (2, 3, 4 months) (3) PCV10: 2 + 1 (2, 4, 9.5 months) (4) PCV10: 2-dose (2, 6 months) (5) PCV13:2 + 1 (2, 4, 9.5 months) (6) Control group: two doses of PCV10 at 18 and 24 months. (7) An additional control group (to evaluate single dose schedule): PCV10 at 24 months	Proportion of infants with IgG concentrations ≥ 0.35 μg/mL did not differ between PCV10 and PCV13 groups at the 10% level for any shared serotype Two doses of PCV13 (Group 5) were non-inferior to 3 doses of PCV10 (Group 2) for 9/10 shared serotypes (excluding 6B) 2 + 1 schedule of PCV10 reduced PCV10-type carriage by 45–62% by 24 months of age, and a 2 + 1 schedule of PCV13 reduced PCV13-type carriage by 36–49% at 12 and 18 months of age. Analyses ongoing for further comparison including B cells and different vaccine schedules	[45,46,47]

RCT: randomised controlled trial; 23vPPV: 23-valent pneumococcal polysaccharide vaccine; ALRI: acute lower respiratory infection; PCV: pneumococcal conjugate vaccine.

**Table 4 vaccines-09-00921-t004:** Pneumococcal vaccine impact studies.

Country	Year Introduced	Vaccine	Study Findings	Reference
Australia (Indigenous Cohort)	2001	PCV7 (3 + 1 PPV)	Elevated risk (~20%) of ALRI requiring hospitalization was observed after each dose of the 7vPCV, compared with that for children (aged 5–23 months old) who did not received 7vPCV. An even greater elevation in risk (39%) was observed in children (aged 5–23 months old) after 23vPPV compared with no receipt of 23vPPV; mostly seen in children who had <3 PCV doses (adjusted HR, 1.81; 95% CI, 1.32–2.48)	[48]
Australia (Indigenous Cohort)	2001	PCV7 (3 + 1 PPV)	Poor completeness of PCV7 schedules within the recommended schedules. Limited evidence that PCV7 reduced the incidence of radiologically confirmed pneumonia among Indigenous infants; no change in all-cause hospitalisation rates or chest x-ray hospitalization. A non-statistically significant declining trend of WHO-defined consolidated pneumonia in vaccinated and non-vaccinated cohorts over time; a non-statistically significant trend towards a vaccine effectiveness of between 16 and 24% following the third dose.	[49]
Australia (Indigenous Cohort)	2001	PCV7 (3 + 1 PPV)	Vaccinated children (born in 2001–2004) had similar rates of otitis media effusion (OME) compared with unvaccinated children (born in 1996–2001) by 6 months of age; time to first OME was not significantly different by group. A lower proportion of vaccinated children experienced tympanic membrane perforation (TMP) in the first 9 months of life, but the proportions were similar by 12 months. Less recurrent TMP; 9% (8/95) versus 22% (11/51) and trends towards reduced incidence of acute otitis media and TMP in the first 2 years of life in vaccinated children compared to unvaccinated.	[50]
Australia	2005	PCV7 (3 + 1 PPV) in 2001 PCV10 (3 + 0) in 2009 PCV13 (3 + 0) in 2011	Pneumonia hospital admission rate for Indigenous Australian children born in the universal PCV period (2005–2012) and younger than 2 years decreased from 23.3/1000 child-years to 15.2/1000 child-years when compared with Indigenous Australian children born in the pre-universal PCV period (2001–2004); non-Indigenous children decreased from 6.7 per 1000 child-years to 4.9 per 1000 child-years Pneumonia hospital admission decreased for vaccinated children (49% reduction; 95% CI 40–55) and unvaccinated children (12% reduction; 95% CI 3–25) younger than 2 years Children born in the universal PCV period (2005-12), unadjusted pneumonia hospital admission rates were significantly lower in children with three or more recorded doses of PCV compared with unvaccinated children	[52]
Australia	2005	PCV7 (3 + 1 PPV) in 2001 PCV10 (3 + 0) in 2009 PCV13 (3 + 0) in 2011	During the universal PCV period, rates of PCV7 serotype-specific IPD rates declined 89.5% reduction among unvaccinated children between the pre-universal and universal PCV7 periods, compared with a 61.4% reduction among vaccinated children; herd immunity Compared with unvaccinated children in the pre-universal period, IPD rates among 3-dose PCV13 recipients were 84% lower in non-Indigenous children and 62% lower in Indigenous children. No statistically significant differences in vaccine efficacy between 1, 2 and 3 doses of PCV13 against IPD due to PCV13 serotypes among non-Indigenous children <2 years old,	[53]
Australia	2009	PCV10 (3 + 0)	In children aged 18-months to <18-years with recurrent protracted bacterial bronchitis, chronic suppurative lung disease or bronchiectasis, receipt of PCV10 was associated with less respiratory symptoms during the follow up (incidence density ratio (IDR) 0.82, 95% CI 0.61, 1.10) and required fewer short-course (<14-days duration) antibiotics (IDR 0.81, 95% CI 0.61, 1.09), compared to no PCV10. Receipt of PCV10 was associated with less hospitalised exacerbations during the follow up (incidence density ratio was 0.15 (95% CI 0.01–0.95) compared to no PCV10	[54]
Australia (Indigenous Cohort)	2001	PCV7 (3 + 1 PPV) in 2001, followed by PCV10 (3 + 1) in 2009 and then PCV13 (3 + 1) in 2011	ALRI rates were lowest in the PCV13 era in association with the significant reduction in bacterial pneumonia hospitalisations compared with the PCV10 (IRR 0.68, 95% CI 0.57–0.81) and PCV7 (0.70, 0.60–0.81) eras. Significant declines for ALRI (–8.1, 95% CI −14.2 to −2.0), all cause pneumonia (−5.5, −8.3 to −2.7), and bacterial pneumonia (–3.4, −5.7 to −1.1) at the transition between the PCV13 and PCV10, but not PCV10 and PCV7 eras.	[55]
Australia	2005	PCV7 (3 + 0) in 2005, followed by PCV10 (3 + 0) in 2009 and then PCV13 (3 + 0) in 2011	Comparing the pre-universal PCV period (1996–2004) with universal PCV periods (2005–2012), the age groups with the greatest declines were children aged 6–11 months (50–63% decline Indigenous; 39–46% decline non-Indigenous) and 2–4 years (54–66% decline Indigenous; 32–44% decline non-Indigenous). Compared to the pre–universal PCV period, pneumococcal pneumonia rates reduced by 60% in non-Indigenous children aged 12–23 months in the universal PCV period; the rates were 4.3 times higher (but not statistically significant) in Indigenous children aged 12–23 months.	[56]
Fiji	2012	PCV10 (3 + 0); 6, 10 and 14 weeks	3 years after introduction: - Vaccine-serotype carriage prevalence declined in 5–8-week-old infants (0.56, 95% CI 0.34–0.93), 12–23-month-olds (0.34, 0.23–0.49) 2–6-year-olds (0.47, 0.34–0.66) and caregivers (0.43, 0.13–1.42) - Carriage of non-PCV10 serotypes increased in Indigenous Fijian infants and children. - Density of PCV10 and non-PCV10 serotypes was lower in vaccinated children than unvaccinated children of the same age group 5 years after introduction: - A reduction in all-cause pneumonia among children aged 24–59 months, but no change among children aged younger than 2 months. - Children aged 2–23 months: ○ 21% (95% CI 5–35) decline for severe or very severe pneumonia ○ 46% (33–56) decline for hypoxic pneumonia ○ 25% (9–38) decline for radiological pneumonia. ○ Mortality reduced by 39% (95% CI 5–62) for all-cause pneumonia, bronchiolitis, and asthma admissions	[31,57]
Laos	2013	PCV13 (3 + 0); 6, 10 and 14 weeks	Two years after introduction: 23% reduction in PCV13-type carriage in children aged 12–23 months, and no significant change in non-PCV13 serotype carriage. Five years after introduction: PCV13 reduced hypoxic pneumonia and pneumonia requiring supplementary oxygen by 37% (95% CI: 6, 57%) in children with pneumonia.	[58,59]
Papua New Guinea	2014	PCV13 (3 + 0); 1, 2 and 3 months	Evaluation ongoing to determine changes in VT pneumococcal carriage in the hospitalised cases, within the community, and indirect effects in the adult age group	[60]
Mongolia	2016	PCV13 (2 + 1); 2, 4 and 9 months	One year after introduction: PCV13 serotype carriage reduced by 52% and 51% in 12–23-month-old and 5–8-week-olds, respectively; increase in non-PCV13 serotype carriage (55%) in 12–23-month-old. Evaluation in children aged 2–59 months ongoing with the following outcomes: hospitalised radiological pneumonia, clinically-defined pneumonia, nasopharyngeal carriage of pneumococci among pneumonia patients and in the community, and severe respiratory infection associated with RSV and/or influenza.	[61,62]

Australian Indigenous population originally receive 3 primary doses of PCV (2, 4, 6 months) + 1 booster dose of PPV at 18 months, while non-Indigenous population receive 3 primary doses of PCV (2, 4, 6 months) with no booster. 23vPPV: 23-valent pneumococcal polysaccharide vaccine; ALRI: acute lower respiratory infection; PCV: pneumococcal conjugate vaccine.

**Table 5 vaccines-09-00921-t005:** Characteristics of HPV VLP vaccines.

Manufacturer	Merck ™ (Gardasil ^®^, 4vHPV)	GlaxoSmithKline ™ (Cervarix ^®^, 2vHPV)	Merck ™ (Gardasil ^®^ 9, 9vHPV)	^ Innovax ™ (Cecolin ^®^, E2vHPV)
L1 VLP types	6, 11, 16 and 18	16 and 18	6, 11, 16, 18, 31, 33, 45, 52 and 58	16 and 18
Dose	20/40/40/20 µg	20/20 µg	30/40/60/40/ 20/20/20/20/20	40/20 µg
Producer cells	*Saccharomyces cerevisiae* (baker’s yeast) expressing L1	*Trichoplusia ni* (Hi 5) insect cell line infected with L1 recombinant baculovirus	*Saccharomyces cerevisiae* (baker’s yeast) expressing L1	*Escherichia coli*
Adjuvant	225 µg aluminium hydroxyphosphate sulfate	500 µg aluminium hydroxide, 50 µg 3-O-deacylated-4′-monophosphoryl lipid A	500 µg aluminium hydroxyphosphate sulfate	Aluminium hydroxyphosphate sulfate
Vaccination schedule	0, 2 and 6 months (15–45 years old) or 0, 6 months (9–14 years old)	0, 1 and 6 months (15–25 years old) or 0, 6 months (9–14 years old)	0, 2 and 6 months (15–45 years old) or 0, 6 months (9–14 years old)	0, 6 months (9–14 years old)

HPV: Human papillomavirus; VLP: Virus-like particle; ^ under review for WHO pre-qualification.

**Table 6 vaccines-09-00921-t006:** HPV vaccine impact studies in Australia.

Study Type	^ Years Post Vaccine Introduction	Cohort	Main Findings
Ecological [75]	3	Women with cervical abnormalities (Victorian Cervical Cytology Registry, *n* ≥ 1 million, pre- and post-vaccination)	Decrease of 0.38% in high-grade cervical abnormalities in women under 18 yo
Cross-sectional [76]	4	Young women (18–24 yo) presenting for cervical cytology screening (Pre-vaccine, *n* = 202; Post-vaccine, *n* = 404)	Lower prevalence of vaccine-type HPV (post- vs. pre-vaccine: 6.7% vs. 28.7%) Lower prevalence of vaccine-type HPV in both vaccinated and unvaccinated women; herd protection Lower prevalence of non-vaccine-type HPV in vaccinated women (post- vs. pre-vaccine 30.8% vs. 37.6%); cross-protection
Observational [85]	5	Women and men at risk of sexually transmitted infections (Sexual Health Services, *n* = 7686 patients (2394 women and 5292 men)	Significant decline in genital warts cases for women under 21 yo (92.6%) and 21–30 yo (72.6%) Significant declines heterosexual men under 21 yo (81.8%) and 21–30 yo (51.1%) No significant decline in wart diagnoses in women and heterosexual men >30 yo
Cross-sectional [82]	5	MSM aged 16–20 yo from sexual health clinics, gay community events, universities, smartphone dating applications, and social networking services (*n* = 400)	Lower prevalence of any quadrivalent vaccine-HPV genotype post vaccination when compared to pre in: - anal (7% vs. 28%); decreased in HPV 6, 11, 16 and 18 - penile (6% vs. 12%); decreased in HPV 6 and 11 - oral (1% vs. 4%)
Cross-sectional [77]	6	Women (18–24 yo) attended Pap screening (Pre-vaccine, *n* = 202; Post-vaccine, *n* = 1058)	Lower prevalence of vaccine-type HPV (post- vs. pre-vaccine: 7% vs. 29.0%) Lower prevalence of vaccine-type HPV in both vaccinated and unvaccinated women; herd protection Decrease in non-vaccine HPV types (HPV 31/33/45)-cross protection in vaccinated women
Observational [86,87]	6	Young women (18–25 yo) recruited through advertisement in Facebook (*n* = 1223)	Very low vaccine targeted HPV genotypes (1.8%, 95% CI: 0.8–2.7%); 11 and 2 cases for HPV 16 and 6, respectively. Prevalence of any of HPV 31/33/45/52/58 genotypes collectively was 6.8% (95% CI: 5.0–8.6%)
Observational [88]	7	Young women (<25 yo) diagnosed with chlamydia (*n* = 1202)	Vaccinated: pre-vaccine vs. post-vaccine HPV prevalence; HPV 6 and 11: 16% vs. 2%, HPV 16 and 18: 30% vs. 4% Unvaccinated: pre-vaccine vs. post-vaccine 4vHPV types prevalence: 41% vs. 19%; herd immunity
Retrospective [89]	7	Women and men diagnosed with genital warts (Sexual Health Clinics, *n* = 4282 (1242 women and 3037 men and 3 transgender)	Overall genital warts decreased from 13.1% to 5.7% between 2004 and 2014 Women <21 yo, genital warts decreased from 18.4% in 2004/2005 to 1.1% in 2013/2014; women >32 yo genital warts increased from 4.0% to 8.5% 45% and 37% reduction in genital warts cases in women and heterosexual men <21 yo; no change for in women and heterosexual men > 32 yo
National data analysis [79]	7	Data registry data from all 8 jurisdictional cervical screening registers, national HPV vaccination register, death index and cancer registers, includes all Australian women aged ≤ 15 who were eligible for vaccine and screened between April 2007 and 31 December 2014.	The adjusted hazard ratio for CIN2/AIS+ was significantly lower for all dose groups compared to unvaccinated women (1 dose 0.65 (95% CI 0.52–0.81), 2 doses 0.61 (0.52–0.72) and 3 doses 0.59 (0.54–0.65). No difference in adjusted hazard ratios for 1 dose and 2 dose recipients when compared to 3 dose recipients
Observational [90]	8	Women with high-grade cervical lesions (18–32 yo)	In 18–25 yo women, the proportion of HPV 16/18-positive CIN3/AIS decreased significantly over time from 69% in 2001–2005 (pre-vaccine), to 62% in 2011–2012 (post-vaccine), to 47% in 2013–2014; no significant change in HPV 16/18 in 26–32-year-olds in 2013–2014 Nonavalent vaccine types accounted for 80% of CIN3/AIS in 18–25 yo women and 90% in 26–32 yo women, 8 years after HPV vaccine introduction.
Cross-sectional [78]	9	Women 18–24 yo and 25–35 yo in 2015 (Pre-vaccine, *n* = 275; Post-vaccine, *n* = 381)	In 18–24 yo: prevalence of 4vHPV types decreased from 22.7% (2005–2007) to 7.3% (2010–2012) and then to 1.5% (2015). In 25–35 yo: prevalence of 4vHPV types decreased from 11.8% (2005–2007) to 1.1% (2015)

^ HPV vaccine introduced in the Australia National Immunisation Program in 2007 (girls-only; in 2013, gender-neutral program). Yo: years old.

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
