# Peer review of "Australia’s Role in Pneumococcal and Human Papillomavirus Vaccine Evaluation in Asia-Pacific"

_vaccines, 2021, doi:10.3390/vaccines9080921_

Round 1

Reviewer 1 Report

Australia´s role in pneumococcal and human papillomavirus vaccine evaluation in Asia-Pacific

The present work is well presented and the authors reviewed and describes the role of Australia in the program and evaluation of pneumococcal (PCV) and human papilloma virus (HPV) vaccines in the Asia-Pacific region.

There are two types of pneumococcal vaccine: PPV-(polysaccharide) (1983) and PCV (Pneumococo conjugate vaccine) (Pneumosil) - PCV10 and PCV13 (2000, 2009, 2010 and 2220).

They described and discussed the diverse pneumococcal vaccine schedules, in different regions (Fiji, Papua New Guinea, indigenous cohort in Australia).

The article also describe the impact studies for pneumococcal vaccines (all types from Australia-indigenous and non-indigenous cohorts from 2001; 2005, 2009, 2012 in Fiji, 2013 in Laos, 2014 in Papua Guinea and 2016 Mongolia.

Author Response

We would like to thank Reviewer 1 for his/her valuable time to review our manuscript. 

Reviewer 2 Report

Vaccines 1320623. Australia’s role in pneumococcal and human papillomavirus 2 vaccine evaluation in Asia-Pacific

Excellent article analysing the many different studies on Pneumococcal and Human Papillomaviruses Vaccine strategies performed by Australian researches.

The article makes an exhaustive review of the many published works on the topics, which can be very helpful in the design of strategies in countries that want to introduce either of both vaccines. However, a cost effectiveness study of the different strategies is missing.

I only have one question: Do the authors consider useful (even necessary) to study the distribution of pneumococcal serotypes causing invasive disease in a country before introduction vaccination schedules? A comment on this might be included in the text.

I would suggest adding the years in which the works collected in tables 2, 3 and 6 were carried out and in page 2, lines 57-58, the years of the estimation of pneumococcal mortality in children.

Finally, it would be desirable, based in the experience and knowledge of the authors, a recommendation of a vaccination schedule for both pathogens in LLMICs.

Author Response

We would like to thank the Reviewer 2 for his/her valuable time and suggestions to improve our manuscript. Please see below for our response to each of the suggestions. We have referred to these changes in the Tracked Changes version of the manuscript.

  1. The article makes an exhaustive review of the many published works on the topics, which can be very helpful in the design of strategies in countries that want to introduce either of both vaccines. However, a cost effectiveness study of the different strategies is missing.

Thank you for this suggestion. While we acknowledge that cost-effectiveness studies are important in informing vaccine implementation, particularly for low- and lower-middle-income countries, the focus of our review was to highlight Australian-led studies related to evaluation of vaccine impact (vaccine efficacy and effectiveness) and alternative vaccine schedules

  1. I only have one question: Do the authors consider useful (even necessary) to study the distribution of pneumococcal serotypes causing invasive disease in a country before introduction vaccination schedules? A comment on this might be included in the text.

Thank you. Yes, this is a very important aspect when considering which vaccine and or schedule to use. We have included the following sentence on Page 5, Line 139-141: “Evaluation of the distribution of pneumococcal serotypes causing invasive diseases and/or carriage prior to vaccine introduction is crucial for measuring vaccine impact and serotype replacement.”

  1. I would suggest adding the years in which the works collected in tables 2, 3 and 6 were carried out and in page 2, lines 57-58, the years of the estimation of pneumococcal mortality in children.

We have included the years when the studies were conducted in Tables 2 and 3. For Table 6, we have included a footnote to indicate the year when HPV vaccine was introduced in Australia (Page 12, line 246-247): “^HPV vaccine introduced in the Australia National Immunisation Program in 2007 (girls-only; in 2013, gender-neutral program).”

On Page 2, Line 57, we have also included the following sentence: “Between 2000-2015, it is estimated….”

  1. Finally, it would be desirable, based in the experience and knowledge of the authors, a recommendation of a vaccination schedule for both pathogens in LLMICs.

We have added the following sentences to the conclusion in Page 14, Line 317-322: “Alternative schedules that alleviate high vaccine costs and logistical constraints in vaccine delivery will improve vaccine access and reduce inequality in LLMICs. Current PCV evidence suggests two primary doses separated by at least two months followed by a later booster dose at or after 9 months of age would provide protection for children in high burden settings, while a single dose HPV vaccine have shown encouraging results. Randomized controlled trials are ongoing to evaluate these schedules and if successful, will significantly improve vaccine access in LLMICs that have yet to introduce these vaccines.”

Reviewer 3 Report

Pneumococcal vaccine and human papillomavirus (HPV) vaccine are two highly successful vaccines in reducing pneumococcal disease and HPV-associated diseases. There are many low- and lower-middle-income countries in the Asia-Pacific region that have not yet introduced pneumococcal vaccine and/or HPV vaccine into their National Immunisation Programs. Toh et al have reviewed the role of Australia in the development and/or evaluation of pneumococcal vaccines and HPV vaccine, including the use of alternative vaccine strategies among countries situated in the Asia-Pacific region. Outcomes of these research programs have had significant global health impact that highlights the importance of these vaccines in preventing pneumococcal disease as well as HPV-associated diseases.

The claims are properly placed in the context of the previous literature. The experimental data support the claims. The manuscript is written clearly enough that most of it is understandable to non-specialists. The authors have provided adequate proof for their claims, without overselling them. The authors have treated the previous literature fairly. The paper offers enough details of methodology so that the experiments could be reproduced.

Minor revisions

Page 9, line 169, “HPV is a broad group of viruses with around 200 genotypes” => “HPV is a broad group of viruses with more than 200 genotypes”

(Of April 2020 there were 228 HPV-types identified).

International Human Papillomavirus Reference Center. Reference clones, Stockholm: Karolinska Institutet.

https://ki.se/en/labmed/international-hpv-reference-center

de Villiers EM. Cross-roads in the classification of papillomaviruses. Virology 2013;445(1-2):2–10. doi: 10.1016/j.virol.2013.04.023.

Doorbar J, Quint W, Banks L, et al. The biology and life-cycle of human papillomaviruses. Vaccine 2012;30(Suppl 5):F55–70. doi: 10.1016/j.vaccine.2012.06.083.

Page 9, line 178, “There are 14 HPV genotypes that are known to cause cancer (oncogenic of high-risk types)” => “There are 20 HPV genotypes that are known to cause cancer (oncogenic of high-risk types)”

(In total 20 HPV genotypes are described causing cervical cancer, Arbyn 2014).

Arbyn M, Tommasino M, Depuydt C, Dillner J. Are 20 human papillomavirus types causing cervical cancer? J Pathol. 2014 Dec;234(4):431-5. doi: 10.1002/path.4424.

https://www.ncbi.nlm.nih.gov/pubmed/25124771

Page 11, Table 6, “Decrease 0.38% in high-grade cervical abnormalities”

I am not impressed by a 0.38 % reduction in high-grade cervical abnormalities. Should it be 38% reduction?

Author Response

Minor revisions

  1. Page 9, line 169, “HPV is a broad group of viruses with around 200 genotypes” => “HPV is a broad group of viruses with more than 200 genotypes”

(Of April 2020 there were 228 HPV-types identified).

International Human Papillomavirus Reference Center. Reference clones, Stockholm: Karolinska Institutet.

https://ki.se/en/labmed/international-hpv-reference-center

de Villiers EM. Cross-roads in the classification of papillomaviruses. Virology 2013;445(1-2):2–10. doi: 10.1016/j.virol.2013.04.023.

Doorbar J, Quint W, Banks L, et al. The biology and life-cycle of human papillomaviruses. Vaccine 2012;30(Suppl 5):F55–70. doi: 10.1016/j.vaccine.2012.06.083.

Thank you. We have reworded our sentence as recommended.

  1. Page 9, line 178, “There are 14 HPV genotypes that are known to cause cancer (oncogenic of high-risk types)” => “There are 20 HPV genotypes that are known to cause cancer (oncogenic of high-risk types)”

(In total 20 HPV genotypes are described causing cervical cancer, Arbyn 2014).

Arbyn M, Tommasino M, Depuydt C, Dillner J. Are 20 human papillomavirus types causing cervical cancer? J Pathol. 2014 Dec;234(4):431-5. doi: 10.1002/path.4424.

Thank you- we have revised the sentence and included the reference (Page 9, Line 180-182).

  1. Page 11, Table 6, “Decrease 0.38% in high-grade cervical abnormalities”

I am not impressed by a 0.38 % reduction in high-grade cervical abnormalities. Should it be 38% reduction?

We have revised the sentence and now reads: “decrease 0.38% in high-grade cervical abnormalities in women under 18 years old”. Cervical cancer incidence peaks between the ages of 30-49 years old in Australia. Cervical cancer takes at least a decade to develop following HPV infection. It is quite remarkable to document any decrease in the incidence of high-grade cervical abnormalities only 3 years following a national HPV vaccination program given to young adolescents aged 12-13 years old in Australia. This study was the first evidence of HPV vaccine impact in Australia. By 8 years following HPV vaccine introduction, the proportion of HPV16/18-positive high-grade cervical abnormalities have decreased significantly from 69% in 2001–2005 (pre-vaccine) to 47% in 2013–2014.